# Reincarnating Reinforcement Learning: Reusing Prior Computation to Accelerate Progress

**Rishabh Agarwal**[1,2*]  **Max Schwarzer**[1,2]
**Pablo Samuel Castro**[1]   **Aaron Courville**[2]   **Marc G. Bellemare**[1,2]
[1] Google Research, Brain Team    [2] MILA

## Abstract

Learning *tabula rasa*, that is without any previously learned knowledge, is the prevalent workflow in reinforcement learning (RL) research. However, RL systems, when applied to large-scale settings, rarely operate tabula rasa. Such large-scale systems undergo multiple design or algorithmic changes during their development cycle and use *ad hoc* approaches for incorporating these changes without re-training from scratch, which would have been prohibitively expensive. Additionally, the inefficiency of deep RL typically excludes researchers without access to industrial-scale resources from tackling computationally-demanding problems. To address these issues, we present *reincarnating* RL as an alternative workflow or class of problem settings, where prior computational work (*e.g.,* learned policies) is reused or transferred between design iterations of an RL agent, or from one RL agent to another. As a step towards enabling reincarnating RL from any agent to any other agent, we focus on the specific setting of efficiently transferring an existing sub-optimal policy to a standalone value-based RL agent. We find that existing approaches fail in this setting and propose a simple algorithm to address their limitations. Equipped with this algorithm, we demonstrate reincarnating RL's gains over tabula rasa RL on Atari 2600 games, a challenging locomotion task, and the real-world problem of navigating stratospheric balloons. Overall, this work argues for an alternative approach to RL research, which we believe could significantly improve real-world RL adoption and help democratize it further. Open-sourced code and trained agents at `agarwl.github.io/reincarnating_rl`.

## 1   Introduction

Reinforcement learning (RL) is a general-purpose paradigm for making data-driven decisions. Due to this generality, the prevailing trend in RL research is to learn systems that can operate efficiently *tabula rasa*, that is without much learned knowledge including prior computational work such as offline datasets or learned policies. However, tabula rasa RL systems are typically the exception rather than the norm for solving large-scale RL problems [4, 13, 55, 75, 85]. Such large-scale RL systems often need to function for long periods of time and continually experience new data; restarting them from scratch may require weeks if not months of computation, and there may be billions of data points to re-process – this makes the tabula rasa approach impractical. For example, the system that plays Dota 2 at a human-like level [13] underwent several months of RL training with continual changes (*e.g.,* in model architecture, environment, *etc*) during its development; this necessitated building upon the previously trained system after such changes to circumvent re-training from scratch, which was done using ***ad hoc*** approaches (described in Section 3).

Current RL research also excludes the majority of researchers outside certain resource-rich labs from tackling complex problems, as doing so often incurs substantial computational and financial cost: AlphaStar [85], which achieves grandmaster level in StarCraft, was trained using TPUs for more than a month and replicating it would cost several million dollars (Appendix A.1). Even the quintessential deep RL benchmark of training an agent on 50+ Atari games [10], with at least 5 runs, requires

---

*Correspondence to Rishabh Agarwal <rishabhagarwal@google.com>.

36th Conference on Neural Information Processing Systems (NeurIPS 2022).

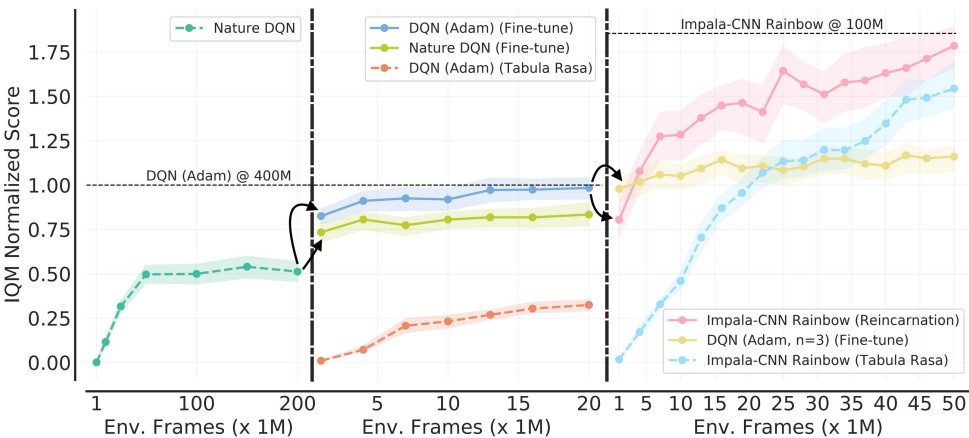

Figure 1: **A reincarnating RL workflow on ALE**. The plots show IQM [2] normalized scores over training, computed using 50 seeds, aggregated across 10 Atari games. The vertical separators correspond to loading network weights and replay buffer for fine-tuning while offline pre-training on replay buffer using QDagger (Section 4.1) for reincarnation. Shaded regions show 95% confidence intervals. We assign a score of 1 to DQN (Adam) trained for 400M frames and 0 to a random agent. **(Panel 1)** *Tabula rasa* Nature DQN [60] nearly converges in performance after training for 200M frames. **(Panel 2) Reincarnation via fine-tuning** Nature DQN with a reduced learning rate leads to 50% higher IQM with only 1M additional frames (leftmost point). Furthermore, fine-tuning Nature DQN while switching from RMSProp to Adam matches the performance of DQN (Adam) trained from scratch for 400M frames, using only 20M frames. **(Panel 3)**. A modern ResNet (Impala-CNN [26]) with a better algorithm (Rainbow [35]) outperforms further fine-tuning $n$-step DQN. **Reincarnating Impala-CNN Rainbow from DQN**, outperforms tabula rasa Impala-CNN Rainbow throughout training and requires only 50M frames to nearly match its performance at 100M frames. See Section 5.

more than 1000 GPU days. As deep RL research move towards more challenging problems, the computational barrier to entry in RL research is likely to further increase.

To address both the computational and sample inefficiencies of tabula rasa RL, we present *reincarnating RL* (RRL) as an alternative research workflow or a class of problems to focus on. RRL seeks to *maximally leverage existing computational work, such as learned network weights and collected data*, to accelerate training across design iterations of an RL agent or when moving from one agent to another. In RRL, agents need not be trained tabula rasa, except for initial forays into new problems. For example, imagine a researcher who has trained an agent $\mathcal{A}_1$ for a long time (*e.g.,* weeks), but now this or another researcher wants to experiment with better architectures or RL algorithms. While the tabula rasa workflow requires re-training another agent from scratch, reincarnating RL provides the more viable option of transferring $\mathcal{A}_1$ to another agent and training this agent further, or simply fine-tuning $\mathcal{A}_1$ (Figure 1). As such, RRL can be viewed as an attempt to provide a formal foundation for the research workflow needed for real-world and large-scale RL models.

Reincarnating RL can democratize research by allowing the broader community to tackle larger-scale and complex RL problems without requiring excessive computational resources. As a consequence, RRL can also help avoid the risk of researchers overfitting to conclusions from small-scale RL problems. Furthermore, RRL can enable a benchmarking paradigm where researchers continually improve and update existing trained agents, especially on problems where improving performance has real-world impact (*e.g.,* balloon navigation [11], chip design [59], tokamak control [24]). Furthermore, a common real-world RL use case will likely be in scenarios where prior computational work is available (*e.g.,* existing deployed RL policies), making RRL important to study. However, beyond some *ad hoc* large-scale reincarnation efforts (Section 3), the community has not focused much on studying reincarnating RL as a research problem in its own right. To this end, this work argues for developing general-purpose RRL approaches as opposed to *ad hoc* solutions.

Different RRL problems can be instantiated depending on how the prior computational work is provided: logged datasets, learned policies, pretrained models, representations, *etc*. As a step towards developing broadly applicable reincarnation approaches, we focus on the specific setting of *policy-to-value* reincarnating RL (PVRL) for efficiently transferring a suboptimal teacher policy to a value-based RL student agent (Section 4). Since it is undesirable to maintain dependency on past teachers for successive reincarnations, we require a PVRL algorithm to "wean" off the teacher

dependence as training progresses. We find that prior approaches, when evaluated for PVRL on the Arcade Learning Environment (ALE) [10], either result in small improvements over the tabula rasa student or exhibit degradation when weaning off the teacher. To address these limitations, we introduce QDagger, which combines Dagger [71] with $n$-step Q-learning, and outperforms prior approaches. Equipped with QDagger, we demonstrate the sample and compute-efficiency gains of reincarnating RL over tabula rasa RL, on ALE, a humanoid locomotion task and the simulated real-world problem of navigating stratospheric balloons [11] (Section 5). Finally, we discuss some considerations in RRL as well as address reproducibility and generalizability concerns.

## 2    Preliminaries

The goal in RL is to maximize the long-term discounted reward in an environment. We model the environment as an MDP, defined as $(\mathcal{S}, \mathcal{A}, R, P, \gamma)$ [69], with a state space $\mathcal{S}$, an action space $\mathcal{A}$, a stochastic reward function $R(s, a)$, transition dynamics $P(s'|s, a)$ and a discount factor $\gamma \in [0, 1)$. A policy $\pi(\cdot|s)$ maps states to a distribution over actions. The Q-value function $Q^\pi(s, a)$ for a policy $\pi(\cdot|s)$ is the expected sum of discounted rewards obtained by executing action $a$ at state $s$ and following $\pi(\cdot|s)$ thereafter. DQN [60] builds on Q-learning [87] and parameterizes the Q-value function, $Q_\theta$, with a neural net with parameters $\theta$ while following an $\epsilon$-greedy policy with respect to $Q_\theta$ for data collection. DQN minimizes the temporal difference (TD) loss, $\mathcal{L}_{TD}(\mathcal{D}_S)$, on transition tuples, $(s, a, r, s')$, sampled from an experience replay buffer $\mathcal{D}_S$ collected during training:

$$\mathcal{L}_{TD}(\mathcal{D}) = \mathbb{E}_{s,a,r,s' \sim \mathcal{D}} \left[ \left( Q_\theta(s, a) - r - \gamma \max_{a'} \bar{Q}_\theta(s', a') \right)^2 \right] \tag{1}$$

where $\bar{Q}_\theta$ is a delayed copy of the same Q-network, referred to as the *target network*. Modern value-based RL agents, such as Rainbow [35], use $n$-*step* returns to further stabilize learning. Specifically, rather than training the Q-value estimate $Q(s_t, a_t)$ on the basis of the single-step temporal difference error $r_t + \gamma \max_{a'} Q(s_{t+1}, a') - Q(s_t, a_t)$, an $n$-step target $\sum_{k=0}^{n-1} \gamma^k r_{t+k} + \gamma^n \max_{a'} Q(s_{t+n}, a') - Q(s_t, a_t)$ is used in the TD loss, with intermediate future rewards stored in the replay $\mathcal{D}$.

## 3    Related work

**Prior *ad hoc* reincarnation efforts**. While several high-profile RL achievements have used reincarnation, it has typically been done in an ad-hoc way and has limited applicability. OpenAI Five [13], which can play Dota 2 at a human-like level, required 10 months of large-scale RL training and went through continual changes in code and environment (*e.g.,* expanding observation spaces) during development. To avoid restarting from scratch after such changes, OpenAI Five used "**surgery**" akin to Net2Net [17] style transformations to convert a trained model to certain bigger architectures with custom weight initializations. AlphaStar [85] employs population-based training (**PBT**) [42], which periodically copies weights of the best performing value-based agents and mutates hyperparameters during training. Although PBT and surgery methods are efficient, they have they *can not* be used for reincarnating RL when switching to arbitrary architectures (*e.g.,* feed-forward to recurrent networks) or from one model class to another (*e.g.,* policy to a value function). Akkaya et al. [4] trained RL policies for several months to manipulate a robot hand for solving Rubik's cube. To do so, they "rarely trained experiments from scratch" but instead initialized new policies, with architectural changes, from previous trained policies using **behavior cloning** via on-policy distillation [20, 67]. AlphaGo [75] also used behavior cloning on human replays for initializing the policy and fine-tuning it further with RL. However, behavior cloning is only applicable for policy to policy transfer and is inadequate for the PVRL setting of transferring a policy to a value function [*e.g.,* 63, 83]. Contrary to such approaches, we apply reincarnation in settings where these approaches are not applicable including transferring a DQN agent to Impala-CNN Rainbow in ALE, and a distributed agent with MLP architecture to a recurrent agent in BLE. Several prior works also **fine-tune** existing agents with deep RL for reducing training time, especially on real-world tasks such as chip floor-planning [59], robotic manipulation [43], aligning language models [6], and compiler optimization [82]. In line with these works, we find that fine-tuning a value-based agent can be an effective reincarnation strategy (Figure 7). However, fine-tuning is often *constrained* to use the same architecture as the agent being fine-tuned. Instead, we focus on reincarnating RL methods that do not have this limitation.

**Leveraging prior computation**. While areas such as offline RL, imitation learning, transfer in RL, continual RL *etc* focus on developing methods to leverage prior computation, such areas don't strive to change how we do RL research by incorporating such methods as a part of our workflow. For completeness, we contrast closely related approaches to PVRL, the RRL setting we study.

- **Leveraging existing agents**. Existing policies have been previously used for improving data collection [14, 16, 28, 77, 89]; we evaluate one such approach, JSRL [83], which improves exploration in goal-reaching RL tasks. However, our PVRL experiments indicate that JSRL performs poorly on ALE. Schmitt et al. [74] propose kickstarting to speed-up actor-critic agents using an interactive teacher policy by combining on-policy distillation [20, 67] with RL. Empirically, we find that kickstarting is a strong baseline for PVRL, however it exhibits unstable behavior without $n$-step returns and underperforms QDagger. PVRL also falls under the framework of agents teaching agents (ATA) [21] with RL-based students and teachers. While ATA approaches, such as action advice [81], emphasize how and when to query the teacher or evaluating the utility of teacher advice, PVRL focuses on sample-efficient transfer and does not impose constraints on querying the teacher. PVRL is also different from prior work on accelerating RL using a heuristic or oracle value function [9, 19, 78], as PVRL only assumes access to a suboptimal policy. Unlike PVRL methods that wean off the teacher, imitation-regularized RL methods [51, 61] stay close to the suboptimal teacher, which can limit the student's performance with continued training (Figure 9).

- **Leveraging prior data**. Learning from demonstrations (LfD) [5, 30, 36, 40, 72] approaches focus on accelerating RL training using demonstrations. Such approaches typically assume access to optimal or near-optimal trajectories, often obtained from human demonstrators, and aim to match the demonstrator's performance. Instead, PVRL focuses on leveraging a suboptimal teacher policy, which can be obtained from any trained RL agent, that we wean off during training. Empirically, we find that DQfD [36], a well-known LfD approach to accelerate deep Q-learning, when applied to PVRL, exhibits severe performance degradation when weaning off the teacher. Rehearsal approaches [62, 66, 76] focus on improving exploration by replaying demonstrations during learning; we find that such approaches are ineffective for leveraging the teacher in PVRL. Offline RL [1, 49, 52] focuses on learning *solely* from fixed datasets while reincarnating RL focuses on leveraging prior information, which can also be presented as offline datasets, for speeding up further learning from environment interactions. Recent work [45, 51, 55, 63] use offline RL to pretrain on prior data and then fine-tune online. We also evaluate this pretraining approach for PVRL and find that it underperforms QDagger, which utilizes the interactive teacher policy in addition to the prior teacher collected data.

## 4 Case Study: Policy to Value Reincarnating RL

While prior large-scale efforts have used a limited form of reincarnating RL (Section 3), it is unclear how to design more broadly applicable RRL approaches. To exemplify the challenges of designing such approaches, we focus on the RRL setting for accelerating training of a student agent given access to a suboptimal teacher policy and some data from it. While a policy-based student can be easily reincarnated in this setting via behavior cloning [*e.g.,* 4], we study the more challenging *policy-to-value* reincarnating RL (**PVRL**) setting for transferring a policy to a value-based student agent. While we can obtain a policy from any RL agent, we chose this setting because value-based RL methods (Q-learning, actor-critic) can leverage off-policy data for better sample efficiency. To be broadly useful for reincarnating agents, a PVRL algorithm should satisfy the following desiderata:

- **Teacher-agnostic**. Reincarnating RL has limited utility if the student is constrained by the teacher's architecture or learning algorithm. Thus, we require the student to be teacher-agnostic.

- **Weaning**. It is undesirable to maintain dependency on past teachers when reincarnation may occur several times over the course of a project, or one project to another. Thus, it is necessary that the student's dependence on the teacher policy can be weaned off, as training progresses.

- **Compute & sample efficient**. Naturally, RRL is only useful if it is computationally cheaper than training from scratch. Thus, it is desirable that the student can recover and possibly improve upon the teacher's performance using fewer environment samples than training tabula rasa.

**PVRL on Atari 2600 games**. Given the above desiderata for PVRL, we now empirically investigate whether existing methods that leverage existing data or agents (see Section 3) suffice for PVRL. The specific methods that we consider were chosen because they are simple to implement, and also because they have been designed with closely related goals in mind.

**Experimental setup**. We conduct experiments on ALE with sticky actions [57]. To reduce the computational cost of our experiments, we use a subset of 10 commonly-used Atari 2600 games: Asterix, Breakout, Space Invaders, Seaquest, Q*Bert, Beam Rider, Enduro, Ms Pacman, Bowling and River Raid. We obtain the teacher policy $\pi_T$ by running DQN [60] with Adam optimizer for 400 million environment frames, requiring 7 days of training per run with Dopamine [15] on P100 GPUs.

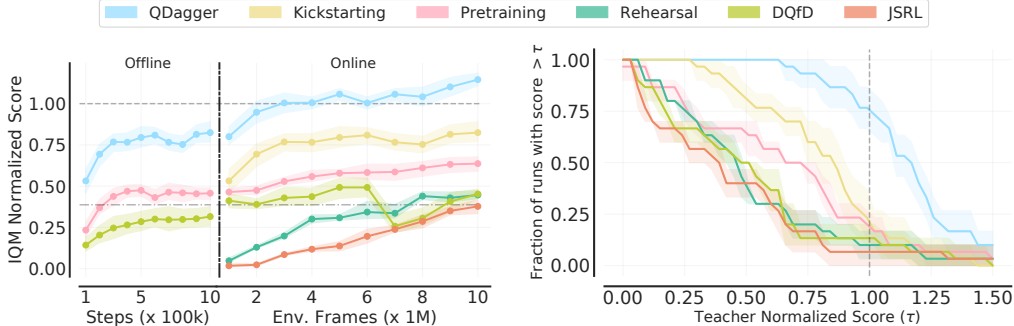

Figure 2: **Comparing PVRL algorithms** for reincarnating a student DQN agent given a teacher policy (with normalized score of 1), obtained from a DQN agent trained for 400M frames (Section 4). Baselines include kickstarting [74], JSRL [83], rehearsal [66], offline pretraining [46] and DQfD [36]. Tabula rasa 3-step DQN student (−· line) obtains an IQM teacher normalized score around 0.39. Shaded regions show 95% CIs. **Left**. Sample efficiency curves based on IQM normalized scores, aggregated across 10 games and 3 runs, over the course of training. Among all algorithms, only QDagger (Section 4.1) surpasses teacher performance within 10 million frames. **Right**. Performance profiles [2] showing the distribution of scores across all 30 runs at the end of training (higher is better). Area under an algorithm's profile is its mean performance while $\tau$ value where it intersects $y = 0.5$ shows its median performance. QDagger outperforms the teacher in 75% of runs.

We also assume access to a dataset $\mathcal{D}_T$ that can be generated by the teacher (see Appendix A.5 for results about dependence on $\mathcal{D}_T$). For this work, $\mathcal{D}_T$ is the final replay buffer (1M transitions) logged by the teacher DQN, which is 100 *times* smaller than the data the teacher was trained on. For a challenging PVRL setting, we use DQN as the student since tabula rasa DQN requires a substantial amount of training to reach the teacher's performance. To emphasize sample-efficient reincarnation, we train this student for only 10 million frames, a 40 *times* smaller sample budget than the teacher. Furthermore, we wean off the teacher at 6 million frames. See Appendix A.3 for more details.

**Evaluation**. Following Agarwal et al. [2], we report interquartile mean normalized scores with 95% confidence intervals (CIs), aggregated across 10 games with 3 seeds each. The normalization is done such that the random policy obtains a score of 0 and the teacher policy $\pi_T$ obtains a score of 1. This differs from typically reported human-normalized scores, as we wanted to highlight the performance differences between the student and the teacher. Next, we describe the approaches we investigate.

- **Rehearsal**: Since the student, in principle, can learn using any off-policy data, we can replay teacher data $\mathcal{D}_T$ along with the student's own interactions during training. Following Paine et al. [66], the student minimizes the TD loss on mini-batches that contain $\rho\%$ of the samples from $\mathcal{D}_T$ and the rest from the student's replay $\mathcal{D}_S$ (different $\rho$ and $n$-step values in Figure A.12).

- **JSRL** (Figure 3, left): JSRL [83] uses an interactive teacher policy as a "guide" to improve exploration and rolls in with the guide for a random number of environment steps. To evaluate JSRL, we vary the maximum number of roll-in steps, $\alpha$, that can be taken by the teacher and sample a random number of roll-in steps between $[0, \alpha]$ every episode. As the student improves, we decay the steps taken by the teacher every iteration (1M frames) by a factor of $\beta$.

- **Offline RL Pretraining**: Given access to teacher data $\mathcal{D}_T$, we can pre-train the student using offline RL. To do so, we use CQL [46], a widely used offline RL algorithm, which jointly minimizes the TD and behavior cloning on logged transitions in $\mathcal{D}_T$ (Equation A.3). Following pretraining, we fine-tune the learned Q-network using TD loss on the student's replay $\mathcal{D}_S$.

- **Kickstarting** (Figure 3, right): Akin to kickstarting [74], we jointly optimize the TD loss with an on-policy distillation loss on the student's self-collected data in $\mathcal{D}_S$. The distillation loss uses the cross-entropy between teacher's policy $\pi_T$ and the student policy $\pi(\cdot|s) = \mathrm{softmax}(Q(s, \cdot)/\tau)$, where $\tau$ corresponds to temperature. To wean off the teacher, we decay the distillation coefficient as training progresses. Note that kickstarting does not pretrain on teacher data.

- **DQfD** (Figure 4, left): Following DQfD [35], we initially pretrain the student on teacher data $D_T$ using a combination of TD loss with a large margin classification loss to imitate the teacher actions (Equation A.4). After pretraining, we train the student on its replay data $\mathcal{D}_S$, again using a combination of TD and margin loss. While DQfD minimizes the margin loss throughout training, we decay the margin loss coefficient during the online phase, akin to kickstarting.

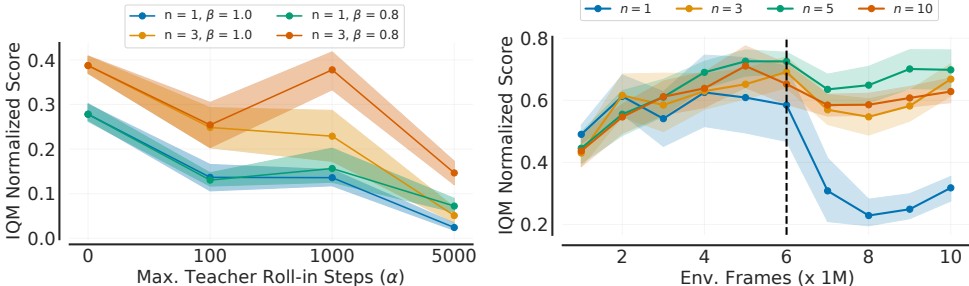

Figure 3: **Left**. **JSRL**. The plot shows teacher normalized scores with 95% CIs, after training for 10M frames, aggregated using IQM across 10 Atari games with 3 seeds each. Each point corresponds to a different experiment, evaluated using 30 seeds, with specific values of JSRL parameters ($\alpha$, $\beta$) and $n$-step returns. **Right**. **Kickstarting**, with different $n$-step returns. The plots show IQM scores over the coures of training. Kickstarting exhibits performance degradation, which is severe with 1-step, and is unable to surpass teacher's performance.

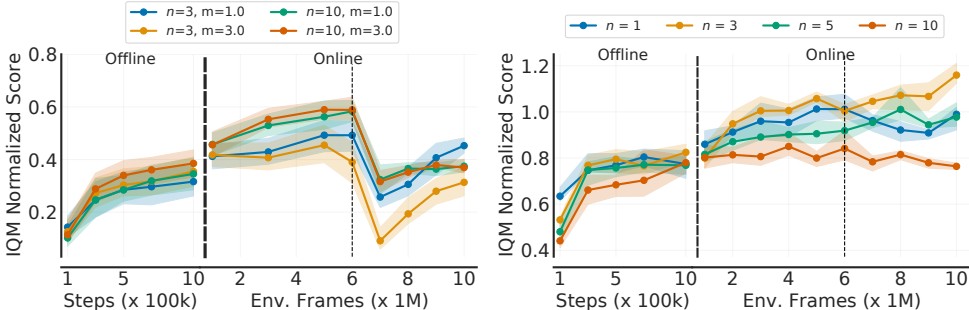

Figure 4: **Left**. **DQfD**. Here, $m$ is the margin loss parameter, which is the loss penalty when the student's action is different from the teacher. **Right**. **QDagger**, with different $n$-step returns. In both, the $1^{st}$ vertical line separates pretraining phase from online phase while the $2^{nd}$ one indicates completely weaning off the teacher.

**Results**. Rehearsal, with best-performing teacher data ratio ($\rho = 1/16$), is marginally better than tabula rasa DQN but significantly underperforms the teacher (Figure 2, teal), which seems related to the difficulty of standard value-based methods to learn from off-policy teacher data [65]. JSRL does not improve performance compared to tabula rasa DQN and even hurts performance with a large number of teacher roll-in steps (Figure 3, left). The ineffectiveness of JSRL on ALE is likely due to the state-distribution mismatch between the student and the teacher, as the student may never visit the states visited by the teacher and as a result, doesn't learn to correct for its previous mistakes [16].

Pretraining with offline RL on logged teacher data recovers around 50% of the teacher's performance and fine-tuning this pretrained Q-function online marginally improves performance (Figure 2, pink). However, fine-tuning degrades performance with 1-step returns, which is more pronounced with higher values of CQL loss coefficient (Figure A.13). We also find that kickstarting exhibits performance degradation (Figure 3, right), which is severe with 1-step returns, once we wean off the teacher policy. Akin to kickstarting, we again observe a severe performance collapse when weaning off the the teacher dependence in DQfD (Figure 4, left), even when using $n$-step returns. We hypothesize that this performance degradation is caused by the inconsistency between Q-values trained using a combination of imitation learning and TD losses, as opposed to only minimizing the TD loss. We also find that using intermediate values of $n$-step returns, such as $n = 3$ (also used by Rainbow [35]), quickly recovers after the performance drop from weaning while larger $n$-step values impede learning, possibly due to stale target Q-values. These results reveal the sensitivity of prior methods in the PVRL setting to specific hyperparameter choices ($n$-step), indicating the need for developing stable PVRL methods that do not fail when weaning off the teacher. For practitioners, the takeaway is to consider this hyperparameter sensitivity when weaning off the teacher for reincarnation.

### 4.1 QDagger: A simple PVRL baseline

To address the limitations of prior approaches, we propose QDagger, a simple method for PVRL that combines Dagger [71], an interactive imitation learning algorithm, with $n$-step Q-learning (Figure 4, right). Specifically, we first pre-train the student on teacher data $\mathcal{D}_T$ by minimizing $\mathcal{L}_{QDagger}(\mathcal{D}_T)$, which combines distillation loss with the TD loss, weighted by a constant $\lambda$. This pretraining phase

helps the student to mimic the teacher's state distribution, akin to the behavior cloning phase in Dagger. After pretraining, we minimize $\mathcal{L}_{QDagger}(\mathcal{D}_S)$ on the student's replay $\mathcal{D}_S$, akin to kickstarting, where the teacher "corrects" the mistakes on the states visited by the student. As opposed to minimizing the Dagger loss indefinitely, QDagger decays the distillation loss coefficient $\lambda_t$ ($\lambda_0 = \lambda$) as training progresses, to satisfy the weaning desiderata for PVRL. Weaning allows QDagger to deviate from the suboptimal teacher policy $\pi_T$, as opposed to being perpetually constrained to stay close to $\pi_T$ (Figure 9). We find that both decaying $\lambda_t$ linearly over training steps or using an affine function of the ratio of student and teacher performance worked well (Appendix A.3). Assuming the student policy $\pi(\cdot|s) = \text{softmax}(Q(s, \cdot)/\tau)$, the QDagger loss is given by:

$$\mathcal{L}_{QDagger}(\mathcal{D}) = \mathcal{L}_{TD}(\mathcal{D}) + \lambda_t \mathbb{E}_{s \sim \mathcal{D}} \Big[ \sum_a \pi_T(a|s) \log \pi(a|s) \Big] \qquad (2)$$

Figure 2 shows that QDagger outperforms prior methods and surpasses the teacher. We remark that DQfD can be viewed as a QDagger ablation that uses a margin loss instead of a distillation loss, while kickstarting as another ablation that does not pretrain on teacher data. Equipped with QDagger, we show how to incorporate PVRL into our workflow and demonstrate its benefits over tabula rasa RL.

## 5 Reincarnating RL as a research workflow

**Revisiting ALE**. As Mnih et al. [60]'s development of Nature DQN established the tabula rasa workflow on ALE, we demonstrate how iterating on ALE agents' design can be significantly accelerated using a reincarnating RL workflow, starting from Nature DQN, in Figure 1. Although Nature DQN used RMSProp, Adam yields better performance than RMSProp [1, 64]. While we can train another DQN agent from scratch with Adam, fine-tuning Nature DQN with Adam and 3-step returns, with a reduced learning rate ( Figure 7), matches the performance of this tabula rasa DQN trained for 400M frames, using a 20 *times* smaller sample budget (Panel 2 in Figure 1). As such, on a P100 GPU, fine-tuning only requires training for a few hours rather than a week needed for tabula rasa RL. Given this fine-tuned DQN, fine-tuning it further results in diminishing returns with additional frames due to being constrained to use the 3-layer convolutional neural network (CNN) with the DQN algorithm.

Let us now consider how one might use a more general reincarnation approach to improve on fine-tuning, by leveraging architectural and algorithmic advances since DQN, without the sample complexity of training from scratch (Panel 3 in Figure 1). Specifically, using QDagger to transfer the fine-tuned DQN, we reincarnate Impala-CNN Rainbow that combines Dopamine Rainbow [35], which incorporates distributional RL [12], prioritized replay [73] and $n$-step returns, with an Impala-CNN architecture [26], a deep ResNet with 15 convolutional layers. Tabula rasa Impala-CNN Rainbow outperforms fine-tuning DQN further within 25M frames. Reincarnated Impala-CNN Rainbow quickly outperforms its teacher policy within 5M frames and maintains superior performance over its tabula rasa counterpart throughout training for 50M frames. To catch up with the performance of this reincarnated agent's performance, the tabula rasa Impala-CNN Rainbow requires additional training for 50M frames (48 hours on a P100 GPU). See Appendix A.4 for more training details. Overall, these results indicate how past research on ALE could have been accelerated by incorporating a reincarnating RL approach to designing agents, instead of always re-training agents from scratch.

**Tackling a challenging control task**. To show how reincarnating RL can enable faster experimentation, we apply PVRL on the *humanoid:run* locomotion task, one of the hardest control problems in DMC [80] due to its large action space (21 degrees of freedom). For this experiment, shown in Figure 5, we use actor-critic agents in Acme [37]. For the teacher policy, we use TD3 [29] trained for 10M environment steps and pick the best run. We find that fine-tuning this TD3 agent degrades performance after 15M environment steps (other learning rates in Appendix A.4), which may be related to capacity loss in value-based RL with prolonged training [47, 56]. For reincarnation, we use single-actor D4PG [8], a distributional RL variant of DDPG [54], with a larger policy and critic architecture than TD3. Reincarnated D4PG performs better than its tabula rasa counterpart for the first 10M environment interactions. Both these agents converge to similar performance, which is likely a limitation of QDagger. This result also raises the question of whether better PVRL methods can lead to reincarnated agents that outperform their tabula rasa counterpart throughout learning. Nevertheless, tabula rasa D4PG requires additional training for 10-12 hours on a V100 GPU to match reincarnated D4PG's performance, which might quickly add up to a substantial savings in compute when running a large set of experiments (*e.g.,* architectural or hyperparameter sweeps).

**Balloon Learning Environment** (BLE) [33]. One of the motivations of our work is to be able to use deep RL in real-world tasks in a data and computationally efficient manner. To this end, the BLE

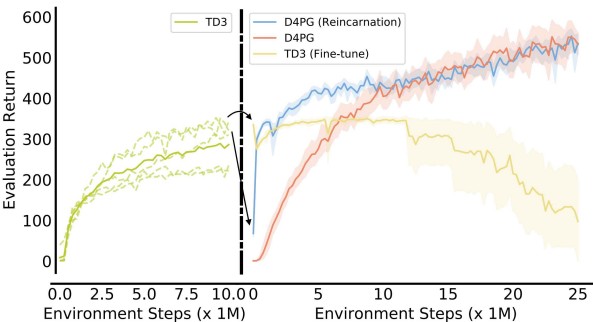
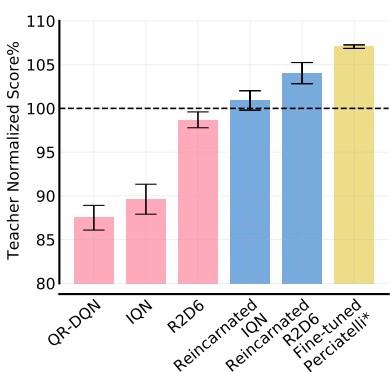

Figure 5: **Reincarnating RL on humanoid:run**. **(Panel 1)**. We observe that TD3 nearly saturates in performance after training for 10M environment steps. The dashed traces show individual runs while the solid line shows the mean return. **(Panel 2)**. Reincarnated D4PG performs better than its tabula rasa counterpart until the first 10M environment steps and then converges to similar performance (with lower variance). Furthermore, training TD3 for a large number of steps eventually results in performance collapse. We use identically parameterized MLP critic and policy networks with 2 hidden layers of size $(256, 256)$ for TD3 but larger networks with 3 hidden layers for D4PG. Shaded regions show 95% CIs based on 10 seeds.

Figure 6: **Comparing BLE agents**. $*$: See main text. We compare QR-DQN [23] with the same MLP architecture as Perciatelli, IQN [22] with DenseNet [39], and R2D6. Reincarnated R2D6 outperforms Perciatelli as well as the tabula rasa agents, but lags behind fine-tuned Perciatelli. We report the mean score (TWR50) across 10,000 evaluation seeds with varying wind difficulty, averaged over 2 independent runs. Error bars show minimum and maximum scores on those runs.

provides a high-fidelity simulator for navigating stratospheric balloons using RL [11]. An agent in BLE can choose from three actions to control the balloon: move up, down, or stay in place. The balloon can only move laterally by "surfing" the winds at its altitude; the winds change over time and vary as the balloon changes position and altitude. Thus, the agent is interacting with a *partially observable* and non-stationary system, rendering this environment quite challenging. For the teacher, we use the QR-DQN agent provided by BLE, called *Perciatelli*, trained using large-scale distributed RL for 40 days on the *production-level* Loon simulator by Bellemare et al. [11] and further fine-tuned in BLE. For our experiments, we train distributed RL agents using Acme with 64 actors for a budget of 50,000 episodes on a single cloud TPU-v2, taking approximately 10-12 hours per run.

In Figure 6, we compare the final performance of distributed agents trained tabula rasa (in pink), with reincarnation (in blue), and fine-tuned (in yellow). We consider three agents, QR-DQN [23] with an MLP architecture (same as Perciatelli), IQN [22] with a Densenet architecture [39], and a recurrent agent R2D6[2] for addressing the partial observability in BLE. When trained tabula rasa, none of these agents are able to match the teacher performance, with the teacher-lookalike QR-DQN agent performing particularly poorly. As R2D6 and IQN have substantial architectural differences from the teacher, we utilize PVRL for transferring the teacher. Reincarnation allows IQN to match and R2D6 to surpass teacher, although both lag behind fine-tuning the teacher. More details in Appendix A.4.2.

When fine-tuning, we are reloading the weights from Perciatelli, which was notably trained on a broader geographical region than BLE and whose training distribution can be considered a *superset* of what is used by the other agents; this is likely the reason that fine-tuning does remarkably well relative to other agents in BLE. Efficiently transferring information in Perciatelli's weights to another agent *without* the replay data from the Loon simulator presents an interesting challenge for future work. Overall, the improved efficiency of reincarnating RL (fine-tuning and PVRL) over tabula rasa RL, as evident on the BLE, could make deep RL more accessible to researchers without access to industrial-scale resources as they can build upon prior computational work, such as model checkpoints, enabling the possible reuse of months of prior computation (*e.g.,* Perciatelli).

## 6 Considerations in Reincarnating RL

**Reincarnation via fine-tuning**. Given access to model weights and replay of a value-based agent, a simple reincarnation strategy is to fine-tune this agent. While naive fine-tuning with the same learning rate ($lr$) as the nearly saturated original agent does not exhibit improvement, fine-tuning

---

[2]R2D6 builds on recurrent replay distributed DQN (R2D2) [44], which uses a LSTM-based policy, and incorporates dueling networks [86], distributional RL [12], DenseNet [39], and double Q-learning [84].

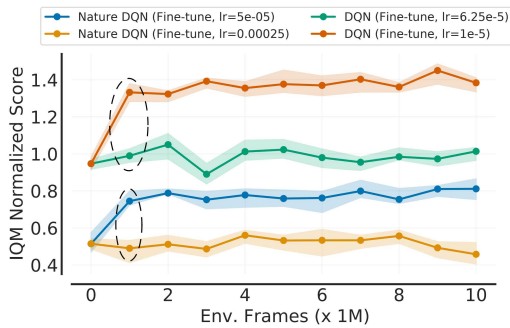
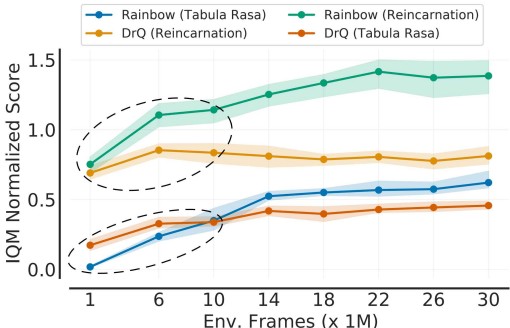

Figure 7: **Reincarnation via fine-tuning** with same and reduced $lr$, relative to the original agent.

Figure 8: **Contrasting benchmarking** results under tabula rasa and PVRL settings.

with a reduced $lr$, for only 1 million additional frames, results in 25% IQM improvement for DQN (Adam) and 50% IQM improvement for Nature DQN trained with RMSProp (Figure 7). As reincarnating RL leverages existing computational work (*e.g.,* model checkpoints), it allows us to easily experiment with such hyperparameter schedules, which can be expensive in the tabula rasa setting. Note that when fine-tuning, one is *forced* to keep the same network architecture; in contrast, reincarnating RL grants flexibility in architecture and algorithmic choices, which can surpass fine-tuning performance (Figures 1 and 5).

**Difference with tabula rasa benchmarking**. Are student agents that are more data-efficient when trained from scratch also better for reincarnating RL? In Figure 8, we answer this question in the negative, indicating the possibility of developing better students for utilizing existing knowledge. Specifically, we compare Dopamine Rainbow [35] and DrQ [90], under tabula rasa and PVRL settings. DrQ outperforms Rainbow in low-data regime when trained from scratch but underperforms Rainbow in the PVRL setting as well as when training longer from scratch. Based on this, we speculate that reincarnating RL comparisons might be more consistent with asymptotic tabula rasa comparisons.

**Reincarnation *vs.* Distillation**. PVRL is different from imitation learning or imitation-regularized RL as it focuses on using an existing policy only as a launchpad for further learning, as opposed to imitating or staying close to it. To contrast these settings, we run two ablations of QDagger for reincarnating Impala-CNN Rainbow given a DQN teacher policy: (1) Dagger [71], which only minimizes the on-policy distillation loss in QDagger, and (2) Dagger + QL, which uses a fixed distillation loss coefficient throughout training (as opposed to QDagger, which decays it; see Equation 2). As shown in Figure 9, Dagger performs similarly to the teacher while Dagger + QL improves over the teacher but quickly saturates in performance. On the contrary, QDagger substantially outperforms these ablations and shows continual improvement with additional environment interactions.

**Dependency on prior work**. While performance in reincarnating RL depend on prior computational work (*e.g.,* teacher policy in PVRL), this is analogous to how fine-tuning results in NLP / computer vision depend on the pretrained models (*e.g.,* using BERT vs GPT-3). To investigate teacher dependence in PVRL, we reincarnate a fixed student from three different DQN teachers (Figure 10). As expected, we observe that a higher performing teacher results in a better performing student. However, reincarnation from two policies with similar performance but obtained from different agents, DQN (Adam) *vs.* a fine-tuned Nature DQN, results in different performance. This suggests that a reincarnated student's performance depends not only on the teacher's performance but also on its behavior. Nevertheless, the ranking of PVRL algorithms remains consistent across these two teacher policies (Figure A.11). See Section 7 for a broader discussion about generalizability.

# 7 Reproducibility, Comparisons and Generalizability in Reincarnating RL

**Scientific Comparisons**. Fairly comparing reincarnation approaches entails using the exactly same computational work and workflow. For example, in the PVRL setting, the same teacher and data should be used when comparing different algorithms, as we do in Section 4. To enable this, it would be beneficial if the researchers can release model checkpoints and the data generated (at least the final replay buffers), in addition to open-source code for their trained RL agents. Indeed, to allow others to use the same reincarnation setup as our work, we have already open-sourced DQN (Adam) agent checkpoints and the final replay buffer at `gs://rl_checkpoints`.

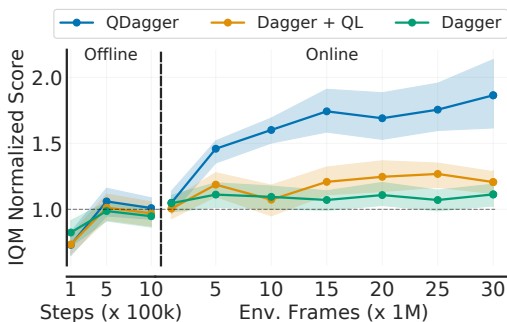

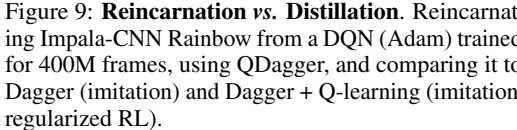

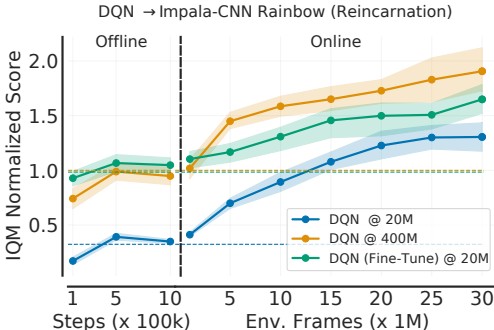

Figure 9: **Reincarnation *vs.* Distillation**. Reincarnating Impala-CNN Rainbow from a DQN (Adam) trained for 400M frames, using QDagger, and comparing it to Dagger (imitation) and Dagger + Q-learning (imitation-regularized RL).

Figure 10: **Reincarnation from different teachers**, namely, a DQN (Adam) policy trained for 20M and 400M frames and fine-tuned Nature DQN in Figure 1 that achieves similar performance to DQN (Adam) trained for 400M frames.

**Generalizability**. The generalizable findings in reincarnating RL would be about comparing algorithmic efficacy given access to existing computational work on a task. As such, the performance ranking of reincarnation algorithms is likely to remain consistent across different teachers. In fact, we empirically verified this for the PVRL setting, where we find that while using two different teacher policies, namely DQN(Adam) *vs.* a fine-tuned Nature DQN, leads to different performance trends but the ranking of PVRL algorithms remain consistent: QDagger > Kickstarting > Pretraining (see Figure 2 and Figure A.11). Practitioners can use the findings from reincarnating RL to try to improve on an existing deployed RL policy (as opposed to being restricted to running tabula rasa RL). For example, this work developed QDagger using ALE and applied it to PVRL on other tasks with existing policies (Humanoid-run and BLE).

**Reproducibility**. Reproducibility *from scratch* is challenging in RRL as it would require details of the generation of the prior computational work (*e.g.,* teacher policies), which may itself has been obtained via reincarnating RL. As reproducibility from scratch involves reproducing existing computational work, it could be more expensive than training tabula rasa, which beats the purpose of doing reincarnation. Furthermore, reproducibility from scratch is also difficult in NLP and computed vision, where existing pretrained models (*e.g.,*, GPT-3) are rarely, if ever, reproduced / re-trained from scratch but almost always used as-is. Despite this difficulty, *pretraining-and-fine-tuning* is a dominant paradigm in NLP and vision [*e.g.,* 18, 25, 34, 38], and we believe that a similar difficulty in RRL should not prevent researchers from investigating and studying this important class of problems. Instead, we expect that RRL research would build on **open-sourced** prior computational work. Akin to NLP and vision, where typically a small set of pretrained models are used in research, we believe that research on developing better reincarnating RL methods can also possibly converge to a small set of open-sourced models / data on a given benchmark, *e.g.,* the agents and data we released on Atari or the 25,000 trained Atari agents released by Gogianu et al. [31], concurrent to this work.

## 8 Conclusion

Our work shows that reincarnating RL is a much computationally efficient research workflow than tabula rasa RL and can help further democratize research. Nevertheless, our results also open several avenues for future work. Particularly, more research is needed for developing better PVRL methods, and extending PVRL to learn from multiple suboptimal teachers [48, 53], and enabling workflows that can incorporate knowledge provided in a form other than a policy, such as pretrained models [41, 79], representations [88], skills [50, 58, 68], or LLMs [3]. Furthermore, we believe that reincarnating RL would be crucial for building embodied agents in open-ended domains [7, 27, 32]. Aligned with this work, there have been calls for collaboratively building and continually improving large pre-trained models in NLP and vision [70]. We hope that this work motivates RL researchers to release computational work (*e.g.,* model checkpoints), which would allow others to directly build on their work. In this regards, we have open-sourced our code and trained agents with their final replay. Furthermore, re-purposing existing benchmarks, akin to how we use ALE in this work, can serve as testbeds for reincarnating RL. As Newton put it "If I have seen further it is by standing on the shoulders of giants", we argue that reincarnating RL can substantially accelerate progress by building on prior computational work, as opposed to always redoing this work from scratch.

## Societal Impacts

Reincarnating RL could positively impact society by reducing the computational burden on researchers and is more environment friendly than tabula rasa RL. For example, reincarnating RL allow researchers to train super-human Atari agents on a single GPU within a span of few hours as opposed to training for a few days. Additionally, reincarnating RL is more accessible to the wider research community, as researchers without sufficient compute resources can build on prior computational work from resource-rich groups, and even improve upon them using limited resources. Furthermore, this democratization could directly improve RL applicability for practical applications, as most businesses that could benefit from RL often cannot afford the expertise to design in-house solutions. However, this democratization could also make it easier to apply RL for potentially harmful applications. Furthermore, reincarnating RL could carry forward the bias or undesirable traits from the previously learned systems. As such, we urge practitioners to be mindful of how RL fits into the wider socio-technical context of its deployment.

## Acknowledgments

We would like to thank David Ha, Evgenii Nikishin, Karol Hausman, Bobak Shahriari, Richard Song, Alex Irpan, Andrey Kurenkov for their valuable feedback on this work. We thank Joshua Greaves for helping us set up RL agents for BLE. We also acknowledge Ted Xiao, Dale Schuurmans, Aleksandra Faust, George Tucker, Rebecca Roelofs, Eugene Brevdo, Pierluca D'Oro, Nathan Rahn, Adrien Ali Taiga, Bogdan Mazoure, Jacob Buckman, Georg Ostrovski and Aviral Kumar for useful discussions.

## Author Contributions

**Rishabh Agarwal** led the project from start-to-finish, defined the scope of the work to focus on policy to value reincarnation, came up with a successful algorithm for PVRL, and performed the literature survey. He designed, implemented and ran most of the experiments on ALE, Humanoid-run and BLE, and wrote the paper.

**Max Schwarzer** helped run DQfD experiments on ALE and as well as setting up some agents for the BLE codebase with Acme, was involved in project discussions and edited the paper. Work done as a student researcher at Google.

**Pablo Samuel Castro** was involved in project discussions, helped in setting up the BLE environment and implemented the initial Acme agents, and helped with paper editing.

**Aaron Courville** advised the project, helped with project direction and provided feedback on writing.

**Marc Bellemare** advised the project, challenged Rishabh to come up with an experimental paradigm in which one continuously improves on an existing agent, and provided feedback on writing.

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
