# OpenReview forum: "Reincarnating Reinforcement Learning: Reusing Prior Computation to Accelerate Progress"
_NeurIPS.cc/2022/Conference — NeurIPS 2022 Accept_

### Official Review · Reviewer_mLNV · 2022-06-30

**Rating:** 8
**Confidence:** 4
**Soundness:** 4 excellent
**Presentation:** 4 excellent
**Contribution:** 4 excellent

**Summary:**

This paper presents an alternative class of problem statements to classical tabula rasa reinforcement learning, that is called reincarnating reinforcement learning (RRL). In RRL, instead of developing algorithms that learn from scratch (tabula rasa) in the RL environment, algorithms can make use of previous policies, and data collected from those policies, to speed up learning. This goes beyond standard offline reinforcement learning as (1) online learning is allowed, and (2) there's a possibility of access to the previous policy, and not just previous data generated from that policy. They instantiate a specific version of RLL in policy-to-value RL (PVRL), where the aim is to increase the speed of value-based online RL methods (such as DQN) with access to a previous (possibly suboptimal) policy and data from that policy. They present Q-dagger, an adjustment to standard deep Q-learning methods which uses n-step returns combined with a Dagger-style loss for both offline training and online from the pretrained policy to improve learning speed, and show it performs better than a variety of strong baselines in this setting in ALE, BLE and humanoid:run.

**Questions:**

Most of my suggestions are discussed in the weaknesses section above.

* Do you see RRL and PVRL as research workflows that will produce (improvements to) algorithms that are useful for tabula rasa RL, or only for RRL/PVRL?
* Could you run an ablation using the Qdagger loss in only the offline pretraining, and then doing standard  and online training stages? From figure 2a it seems that a lot of the benefit of Qdagger comes from the offline pretraining stage - it would be interesting to see how much additional benefit (on top of standard Q-learning) is gained by using the Qdagger loss during online training.

**Limitations:**

I believe the limitations are adequately addressed, given the fixes recommended in the Weaknesses section (which are more focused on unclear description of the framework that I think leads implicitly to unstated limitations in generality).

**Strengths And Weaknesses:**

# Strengths

The paper is well-written and clear to read. While (as mentioned in the paper) this kind of problem setting has been tackled before, it has always been in an ad-hoc way, and a more thorough motivation and investigation of it is beneficial and original. Further, it's likely that the vast majority of the usage RL algorithms in real-world settings will be in scenarios where previous data and/or policies are available, which makes this setting an important one to study.

The PVRL problem/solution setting is interesting, and their algorithm is robustly demonstrated to perform better than existing solutions on a range of benchmarks.

For all these reasons I'm recommending acceptance, but I think there are several issues, in terms of framing of the problem statement and contribution, that would make the paper clearer and aid future work more easily (see Weaknesses below).

# Weaknesses

## Framing

While the RRL class of problem settings in an important and understudied one, I think the framing of the paper slightly confused what exactly RRL __is__. RRL seems like a class of problem settings, of which offline RL is one (in the limit of no online interaction or policy access), and PVRL is another (in which access to a previous policy and data from that policy is assumed). It would be beneficial to describe other possible instantiations of this problem setting; some obvious ones are having access to a previous policy but no data (which is often the setting in kickstarting-style methods), or conversely data but no policy (which is offline RL, possibly plus online fine-tuning); other factors of variation are the quality of the policy and/or data, and the amount of online training allowed.

I think these settings are all valuable, and it's not necessarily the case that one is more general or flexible than another (as claimed for example on line 35; PVRL requires both a fixed dataset and previous policy, making it less general than methods that only require one or the other). It would also make certain distinctions clear that I think aren't in the paper; for example in line 165 it's claimed that assuming access to a previous policy allows one to assume access to a dataset from that teacher "without loss of generality", which doesn't seem to be the case, as there are examples where we have a previous policy but not necessarily data from that policy (we could only have data it was trained with, or no data at all).

To be specific, in the PVRL case, I think it makes more sense to just consider the constraints on the problem setting (previous policy and data, online learning allowed), and not couple that with the type of algorithm being trained (in this case a value-based method). There are many possible solutions to this problem, some of which are based on value-based RL methods and some are not. Hence I'd recommend defining (something like) PDRRL as policy-and-data-RRL as a problem instance within RRL, with PVRL as a class of solutions to this problem.

## Reproducibility

I think it's also important to address the reproducibility and comparability concerns of this research workflow or class of problems. For example consider the passage in lines 47-52:

> For example, imagine a researcher who has trained a deep RL agent A1 for a long time (e.g., weeks), but now this or another researcher wants to experiment with better algorithms or architectures. While the tabula rasa workflow requires re-training another agent from scratch, reincarnating RL provides the more viable option of transferring A1 to another agent and training this agent further, or simply fine-tuning A1 (Figure 1).

While this saves computation for the researcher, to make sure their algorithm is fairly compared with others in the same setting, other algorithms need to be trained with the same flow, which might require even more compute (if we need to reproduce the week-long training as well as the faster fine-tuning). As such, it would be beneficial (alongside releasing the code and model checkpoints) to release the data generated from those models - this means future researcher can work in exactly the same setting (i.e. the same inputs to their PVRL algorithm) so that the comparison is fair. Problems of this nature are even raised in the **Dependency on prior work** paragraph in section 6.

## Concrete recommendations

* Adjust the abstract and intro to frame RRL as a class of problems, of which PDRRL is one instance. Describe some other relevant/significant problem instances
* Adjust the framing of PVRL to be a class of solutions to the PDRRL problem
* Adjust the claims in L35 and L165, in light of the new framing above.
* Address the reproducibility/comparability concerns; specifically, if another researcher was to attempt to produce a new method for PVRL, how would they go about doing that, to make sure their algorithm's results are comparable with those reported here.

If these recommendations and weakensses are addressed (which I think is only a matter of some light rewriting of the paper) then I'm happy to recommend a strong accept.

EDIT: I thank the authors for their detailed response and the adjustments they've made. Given that these addressed my concerns, I'm happy to raise my score to a strong accept (8).

---

> ### Author Response · Authors · 2022-08-01
> **Author Response (Part 1): Incorporated the recommendations, added a discussion on reproducibility**
>
> We thank the reviewer for their constructive feedback. We have revised the paper, with $\textcolor{red}{changes\ in\ red}$, to incorporate their recommendations:
>
> - We acknowledge that reproducibility and comparability is an important concern and have added a detailed discussion in **Appendix A.2**.
>
> - Modified the abstract and intro to frame RRL as a class of problem settings and described some possible instantiations in Appendix A.6.
>
> - Changed the framing of PVRL to be an instance of the RRL problem given access to policy and some data from it. (Lines 143-145).
>
> - Adjusted the claims in L35 and L165 (now L139 and L167).
>
> - Added experiments & discussion in **Appendix A.7** about applying QDagger without access to teacher’s replay buffer (Figure A.19) and impact of QDagger loss in online RL phase (Figure A.20).
>
> > **It's also important to address the reproducibility and comparability concerns of this research workflow or class of problems.**
>
> **Comparability**. We totally agree that the same prior computational work should be utilized for fairly comparing reincarnation approaches. Thus, it would be beneficial to release both the model checkpoints and the data generated (at least the final replay buffers). Indeed, to allow others to use the reincarnation setup in our work, we have already open-sourced [DQN model checkpoints and replay buffer](https://console.cloud.google.com/storage/browser/rl_checkpoints) in the original submission (Appendix A.2).
>
> **Reproducibility**. As *reproducibility from scratch* involves reproducing existing computational work, which could be more expensive than training tabula rasa, it beats the purpose of doing reincarnation. Furthermore, *reproducibility from scratch* is also difficult in NLP and computed vision, where existing pretrained models (e.g., GPT-3) are rarely, if ever, reproduced / re-trained from scratch but almost always used as-is. Despite the difficulty of reproducibility from scratch, pretraining-and-fine-tuning is a dominant paradigm in NLP and vision, and we believe that a similar challenge in reincarnating RL should not prevent researchers from investigating and studying this important problem.
>
> Instead, we expect that research in reincarnating RL would build on open-sourced prior computational work. Akin to NLP and computer vision, where typically a small set of pretrained models are used in research, we believe that for reincarnating RL research can also possibly converge to a small set of open-sourced models / data on a given benchmark, for example, the teacher DQN agents and data we released on Atari.
>
> > **All RRL settings are all valuable, and it's not necessarily the case that one is more general or flexible than another (as claimed for example on line 35; PVRL requires both a fixed dataset and previous policy, making it less general than methods that only require one or the other).**
>
> Indeed, we agree that all RRL settings are valuable and we didn’t mean to imply one setting is more general than other. We have rephrased the sentence to be in L135 (now L139) to indicate that when we have access to an interactive policy in addition to the dataset, this policy can be queried during student’s training.
>
> > **In line 165, it's claimed that assuming access to a previous policy allows one to assume access to a dataset from that teacher "without loss of generality", which doesn't seem to be the case, as there are examples where we have a previous policy but not necessarily data from that policy.**
>
> We have removed the “without loss of generality” (L167). Furthermore, we added new results showing how we can use QDagger without any access to an offline dataset by generating data using the teacher policy and using that for QDagger pre-training.
>
> Specifically, we ran a QDagger ablation on 4 games where we do not assume access to the teacher's replay buffer. As shown in Figure A.19, we found that collecting the same amount of teacher data as the offline replay buffer leads to comparable performance on 3/4 games (Asterix, Qbert, Seaquest). Thus, QDagger can be used successfully given access to only the pretrained policy.
>
> > **In the PVRL case, I think it makes more sense to just consider the constraints on the problem setting and not couple that with the type of algorithm being trained (in this case a value-based method).**
>
> Indeed, we have done this reformulation in Lines 143-145. We have also highlighted that we focus on the PVRL setting because a policy-based student can be easily reincarnated in the PDDRL setting via behavior cloning, it was unclear how to reincarnate a deep Q-learning agent in this setting.

---

> > ### Author Response · Authors · 2022-08-01
> > **Author response (Part 2): Addressing other questions**
> >
> > > **Do you see RRL and PVRL as research workflows that will produce (improvements to) algorithms that are useful for tabula rasa RL, or only for RRL/PVRL?**
> >
> > Extrapolating our results in Figure 8 that sample-efficient tabula rasa algorithms may not be sample-efficient for RRL / PVRL, it is possible that vice versa would be true too. If we are interested in algorithms that perform well both in tabula rasa and RRL settings, they should be benchmarked in both settings. Currently, we mainly benchmark algorithms tabula rasa, making it unclear whether they’d work for reincarnation.
> >
> > > **Could you run an ablation using the Qdagger loss in only the offline pretraining, and then doing standard and online training stages? ..  would be interesting to see how much additional benefit (on top of standard Q-learning) is gained by using the Qdagger loss during online training.**
> >
> > We ran this ablation on 4 games and added the results in Appendix A.7. As shown in Figure A.20, we see that the online phase helps improve performance on some games (significant improvement in Space Invaders, while small improvement in Asterix) while not having much effect on others.
> >
> > On the more complex BLE domain, we only had access to a small amount of replay data relative to the training budget of the Perciatelli teacher. Our preliminary experiments on BLE indicated that using the QDagger loss only during the offline phase resulted in much lower performance than using QDagger for both offline and online phase
> >
> >
> >
> > -------------------------------------------------
> > *We hope that most of the weaknesses pointed by the reviewer have been addressed and, if so, they would reconsider their assessment. We’d be happy to engage in further discussions.*

---

> > > ### Comment · Reviewer_mLNV · 2022-08-05
> > > **Concerns addressed**
> > >
> > > I thank the authors for their detailed response and the adjustments they've made. Given that these addressed my concerns, I'm happy to raise my score to a strong accept (8).

---

### Official Review · Reviewer_VM1W · 2022-07-11

**Rating:** 6
**Confidence:** 4
**Soundness:** 3 good
**Presentation:** 3 good
**Contribution:** 3 good

**Summary:**

This paper proposes a new approach for reincarnating RL agents from pre-trained policies with different architectures and argues that RL papers should also adopt 'pre-training and fine-tuning' approach instead of training everything from scratch. In order to transfer pre-trained policy with different architectures, the method proposes Q-Dagger that learns value functions from a sub-optimal teacher policy and that is capable of continually decreasing the dependency on the sub-optimal teacher. Proposed method is compared to various baseline schemes on Atari and Balloon learning environments.


**Questions:**

- It would be nice to add a discussion on the dependency of the proposed method on the availability of offline datasets. Or I'm open to any follow-up discussion whether it's a limitation or not.
- More experimental details on BLE would be nice for self-containedness of the paper
- More clear explanation and take-away message of n-step experiemnts would be nice.
- Clear explanation about TD3 performance degradation on humanoid run, or additional experiments with SAC without such phenomenon could make experimental results more convincing.

**Limitations:**

Limitation is not clearly discussed in the paper but potential negative societal impact is well discussed.

**Strengths And Weaknesses:**

Strengths
- Idea of reincarnating RL policies is definitely should be of interest to various researchers.
- Idea is simple and working well
- Exhaustive experiments with many baselines

Weaknesses
- Transferring pre-trained policy is definitely important and the paper proposes an intuitive and effective method for doing so. However, the proposed method is still limited in that it quite heavily depends on the availability of offline datasets, which have size of ~1M samples. This assumption gets often more difficult to hold because usually a replay buffer is implemented as a queue of fixed size in memory instead of storing all transitions in storage. The main benefit of pre-training and fine-tuning scheme in CV and NLP domains is that we can fine-tune a pre-trained model on datasets from target domains, which has a much smaller size than large, often huge, pre-training datasets. In this point of view, it is a bit difficult to see that the proposed method can be a generic solution for reincarnating RL and it's not clear whether only releasing pre-trained checkpoints could enable us to obtain the benefit as stated in the paper. It would be much more nice if the paper can show that the method works without pre-training student on offline dataset. I would not say that this experimental results are necessary because the method and results are still interesting, but more discussion on this front would make paper more interesting and insightful for future researches.
- The paper is a bit not self-contained because it's missing some experimental details in Balloon Learning Environment, which is quite new benchmark so that it makes a bit difficult to understand the details. For instance, it is difficult to grasp what is Perciatelli and why is it called Perciatelli, and it's even not in the referred paper.
- Experimental results on n-step returns are quite confusing; what's the main point of n-step return experiments? I understand that some methods are unstable without n-step returns, but why it's so important and what's the conclusion and practical take-away?
- TD3 performance on humanoid run is also quite confusing; what is the reason for performance decrease of TD3? Does this also happen when we train TD3 from scratch without pausing the training after 10M steps? If so, why does this happen? and what is the reason TD3 is chosen for solving the task? It is known that SAC can solve humanoid run task without such degradation (see https://github.com/denisyarats/pytorch_sac for performance of SAC on humanoid run). This makes results a bit less convincing on humanoid experiments.

---

> ### Author Response · Authors · 2022-08-01
> **Author Response: Added discussion for QDagger without offline dataset, takeaway from n-step experiments and BLE details**
>
> We thank the reviewer for their constructive feedback. We have made the following changes, $\textcolor{red}{shown\ in\ red}$, in the revision to address their concerns:
>
> - Added results in **Appendix A.7** showing that QDagger works reasonably well without an existing offline dataset (teacher's last replay buffer).
>
> - More experimental details about BLE in **Appendix A.5.2** for self-containedness.
>
> - Added takeaway from n-step results (**L221 - 224**).
>
> - Included additional clarifications about humanoid:run experiments in Appendix A.5.1.
>
> > **The proposed method is limited in that it depends on the availability of offline datasets, which have size of ~1M samples. This assumption gets difficult to hold because a replay buffer is a queue of fixed size instead of storing all transitions.**
>
> We’d like to clarify that the replay buffer for value-based Atari agents is typically of size 1M transitions, which is 100 times smaller than the data seen by the teacher DQN agent trained for 400M frames (4 frames = 1 transition in Atari). This is somewhat analogous to using a pre-trained agent on large amounts of data (400M frames) and reincarnating another agent given access to a much smaller offline replay dataset (1M transitions) and environment interactions (10M frames).
>
> > **It would be nice to add a discussion on the dependency of the proposed method on the availability of offline datasets.**
>
> If we don’t have access to the teacher's last replay buffer (i.e., the offline dataset), we can simply generate data using the teacher policy during the online RL phase (this would cost environment samples) and use this teacher collected data for pre-training with QDagger loss (analogous to the behavior cloning phase in Dagger).
>
> To verify this empirically, we ran a QDagger ablation on 4 games where we do not assume access to the teacher's replay buffer. As shown in **Figure A.19**, we found that collecting the same amount of teacher data as the offline replay buffer leads to comparable performance on 3/4 games (Asterix, Qbert, Seaquest). Thus, QDagger works reasonably well given access to only the pretrained policy.
>
> > **Experimental results on n-step returns are confusing; what's the main point of n-step experiments?**
>
> We agree with the reviewer that the n-step results can be overwhelming and have revised the paper to include a clear takeaway.
>
> The degradation in performance without n-step returns (e.g., kickstarting) and even with n-step returns (e.g., DQfD) reveals the sensitivity of prior methods to a specific hyperparameter choice in the PVRL setting. For researchers, this indicates the need for developing less hyperparameter-sensitive PVRL methods that do not fail when weaning off the teacher. For practitioners, the takeaway is to consider this hyperparameter sensitivity when weaning off the teacher for reincarnation.
>
> > **The paper is a bit not self-contained .. missing experimental details in BLE.**
>
> We acknowledge this oversight and have added more experimental details on BLE about the station keeping problem, evaluation, environment, teacher and outlined the details about hyperparameters and architectures in *Appendix A.5.2*.
>
> > **What is Perciatelli?**
>
> We’ve rephrased the lines 284-285 to clarify this. Perciatelli is the name of the RL agent (QR-DQN with a MLP architecture) trained by Bellemare et al in their Nature paper using a distributed RL setup and was released with BLE as a starter agent.
>
> > **TD3 performance on humanoid run is also quite confusing .. makes results a less convincing on humanoid.**
>
> We’d like to clarify that the purpose of humanoid:run experiments is to show the utility of reincarnation in a complex continuous control environment given access to a pre-trained policy. Irrespective of fine-tuning TD3’s performance, we can show the benefits of using reincarnation over tabula rasa D4PG. For example, the reincarnated D4PG achieves [SAC’s performance](https://github.com/denisyarats/pytorch_sac) after 10M frames (score of ~400) in only half the number of samples (i.e., 5M frames).
>
> > **Reason for performance decrease of TD3 -- happens for tabula rasa?**
>
> The degradation in performance also happens with tabula rasa TD3 and different learning rates for fine-tuning (Figure A.16) but only after training for 25M frames. We hypothesize that this degradation is likely caused by loss in network’s capacity with prolonged training in value-based RL [[Kumar et al.](https://arxiv.org/abs/2010.14498),  [Lyle et. al](https://openreview.net/forum?id=ZkC8wKoLbQ7)].  Further investigation of this degradation is outside the scope of the work. We’d like to emphasize that TD’s degradation does not affect our conclusions about the efficacy of reincarnating RL over a tabula rasa workflow.
>
> ----------------------------------
> *We would appreciate if the reviewer can confirm that most of their concerns have been addressed and, if so, reconsider their assessment. We’d be happy to engage in further discussions.*

---

> > ### Comment · Reviewer_VM1W · 2022-08-07
> > **Response**
> >
> > Thanks for the detailed response to my review and also other reviewers from different reviewers.
> >
> > - I think the modification made in the rebuttal -- toning down on some strong claims about the problem setup (as pointed out by mLNV) -- could definitely clarify the position of the work in terms of different works. A comment I have on this front is missing related works or references for A.6, especially on leveraging the pre-trained representations from images and videos given the recent surge of relevant works in the field. Currently it's missing all the references (not only for representation pretraining) in A.6.
> >
> > - I still have a concern on the dependency on the dataset. It could be 'simply collecting 1M samples' for simulated environment, especially game environments like Atari, but it's a quite big assumption on the scenario where collecting samples could be costly. If the pre-trained policy is good enough to collect samples without safety issue or incurring significant cost, that would mean the pre-trained policy is already of high-quality which can be deployed to complex environments; then we have to think of whether we really would need fine-tuning or re-incarnating the pre-trained policy. I would not say that we have to completely rid of the online interaction, because we would need online interaction anyway for re-training the policy, but the dependency on the dataset consisting of tens or hundreds of episodes only for value learning seems a bit strong assumption to me.
> >
> > Due to the aforementioned limitation of the method, I would like to maintain the current score of weak accept.

---

> > > ### Author Response · Authors · 2022-08-07
> > > **Follow-up response & clarification**
> > >
> > > We thank the reviewer for reading our response and engaging in follow-up discussion.
> > >
> > > >  **If the pre-trained policy is good enough to collect samples without safety issue or incurring significant cost, that would mean the pre-trained policy is already of high-quality which can be deployed to complex environments; then we have to think of why we have to consider fine-tuning or re-incarnating the policy.**
> > >
> > > While the pre-trained policy achieves good performance, it is often desirable to further improve this policy's performance (via architectural / algorithmic changes). This is especially relevant on real-world RL settings where improving performance leads to real-world impact, such as chip design, tokamak control etc but tabula rasa re-training is computationally / data expensive. The BLE domain shows this use-case: given access to Perciatelli trained via distributed RL training for more than a month, we show how fine-tuning / reincarnation can improve upon this policy compared to tabula rasa with a finite compute/sample budget.
> > >
> > > Furthermore, on Atari, we show that we can improve upon the teacher policy, DQN(Adam) policy trained for 400M frames, by switching to an Impala architecture and Rainbow algorithm and obtain nearly *2x higher IQM* performance than the teacher (Figure 9).
> > >
> > > > **It could be 'simply collecting 1M samples' for simulated environment, especially game environments like Atari, but it's a quite big assumption on the scenario where collecting samples could be costly.**
> > >
> > > Indeed, PVRL methods that perform well with smaller number of environment samples would be desirable. Since most prior methods were not designed for policy to value reincarnating RL (PVRL) due to lack of research in this setting, this work is arguing for doing research in PVRL, where we intend for QDagger to serve as a starting point.
> > >
> > > >  **I would not say that we have to completely rid of the online interaction, because we would need online interaction anyway for re-training the policy, but the dependency on the dataset consisting of tens or hundreds of episodes only for value learning seems a bit strong assumption to me.**
> > >
> > > Collecting data from teacher instead of the student is simply a way of how we choose to use the online interactions in the environment. For a given budget of online interactions, we can choose to use some part of the budget for teacher data collection while remaining budget for student data collection (akin to Figure A.19). Furthermore, not using teacher's final replay buffer or teacher collected data for pretraining corresponds to kickstarting, which results in poor performance relative to QDagger.
> > >
> > >
> > >
> > > >  **Missing related works or references for A.6, especially on leveraging the pre-trained representations from images and videos given the recent surge of relevant works in the field. Currently it's missing all the references (not only for representation pretraining) in A.6.**
> > >
> > > For leveraging pre-trained representations, a large fraction of work is about how to learn these representations (such as in unsupervised RL) and we'll add citations of representative work. For other examples (such as leveraging existing agents / data), the related work discusses those papers and we'll also include them in Appendix A.6 for completeness.

---

### Official Review · Reviewer_jTCn · 2022-07-12

**Rating:** 7
**Confidence:** 2
**Soundness:** 2 fair
**Presentation:** 3 good
**Contribution:** 2 fair

**Summary:**

The authors present the concept of "reincarnating RL" as an alternative to "learning tabular rasa". Reincarnating RL is very generally about using prior data/computation to more efficiently train new agents. They then introduce an algorithm called "QDagger" which is meant to solve a specific instance of "reincarnating RL" called "policy-to-value reincarnating RL" (PVRL). It performs well relative to baselines.

The main contributions of the paper seem to be:

1. The promotion of "reincarnating RL" as an alternative approach to "learning tabular rasa"; along with examples of it as a research workflow.

2. QDagger as a solution to a specific instance of "reincarnating RL" (PVRL).

I will be writing the review considering both of these contributions separately.

**Questions:**

1. What example settings (beyond BLE) are you thinking of when it comes to "real-world RL adoption"? In what real-world scenarios would a broader community improving existing trained agents be helpful? How do you imagine this looks in practice?

2. How is a real-world practitioner supposed to used the learnings from this line of work on a new task? Suppose that a series of papers for reincarnating RL that build on each other demonstrate SOTA performance on some set of tasks. Is the real-world practitioner supposed to then go through each of the different training / reincarnation phases? Right now, it is easy to run tabula rasa RL on a new environment.

**Limitations:**

This paradigm introduces its own host of problems around reproducibility and generalization. As mentioned in the paper, "reincarnation from two distinct policies that perform similarly results in different performance trends" (Line 336). Thus, it seems as though it may be very easy to overfit to specific implementations and seeds of the pre-trained agents.

Furthermore, the precise details of the generation of the teacher policy / data (which themselves may have been reincarnated) become important for reproducibility; adding on to the challenge of reproducing work from scratch. I do not believe that this is addressed in the paper.



**Strengths And Weaknesses:**

# Originality:

## On "Reincarnating RL":

Strength: This is an interesting notion that has not been explicitly discussed much in the literature before.

Weakness: There is plenty of work that would fall into this vague category, as seen in the "Related Work" section. While this umbrella term / notion has not been explicitly discussed much, I'm not sure if generically promoting (and also coining a term for) a category of works could be considered an original contribution.

## On "QDagger":

Strength: The specific setting has not been previously studied much, and the precise details of this algorithm for this specific setting have not been proposed before.

Weakness: This does not seem to be particularly original. The authors write that "kickstarting [can be viewed] as another ablation that does not pretrain on teacher data" (Line 236). Indeed when looking at the algorithms, they are very similar. If QDagger is just "kickstarting with pretraining" then it is not very original.

# Quality:

## On "Reincarnating RL":

I'm really not sure how to evaluate this contribution in terms of quality.

Strength:

The authors make a coherent argument that more attention in this area would better democratize and improve real-world RL adoption. They also show through case studies that experimenting under this paradigm is computationally cheaper and more accessible. Promoting the release of model checkpoints is helpful to the research community.

Weakness:

1. I'm not sure if NeurIPS is the correct venue for what seems to largely be posturing / call-to-action. This interpretation may seem uncharitable, but significant portions of the paper are dedicated to this broad discussion instead of experiments, analysis, or theory.

2. I'm not sure if I agree with "reincarnating RL" as being a superior approach for democratizing RL research. On the whole, it asks the question of what researchers should care about in reinforcement learning. A large part of the field is about studying and understanding how intelligent agents might/should learn tabula rasa (which is valuable in and of itself), not as much about real-world RL adoption. For example, the breakthrough DQN results in Atari are not interesting because they achieve high scores, but rather because a single algorithm could solve a large portion of the games tabula rasa. Only in the cases in which we "care" about ultimate performance (e.g. DOTA, Starcraft, Rubik's cube -- note that these are also not particularly useful in the real-world) do we consider reincarnating RL. Such scenarios currently seem rare in the real-world -- and it's unclear if this is because of a lack of work in reincarnating RL. On the other hand, environments such as [Minatar](https://github.com/kenjyoung/MinAtar) democratize research and allow for cheaper testing of tabula rasa RL.

3. See the limitations section on reproducibility. This is problematic since science is about repeatably testable hypotheses. While it would be neat for a broad community to continually improve existing trained agents, it would not be a scientific endeavor as much as an engineering one.

## On "QDagger":

Strengths: The experiments and setup were well-done.

Weaknesses: N/A.

# Clarity:

Strength: The paper as a whole is very well-written and clear.

Weakness: N/A.

# Significance:

## On "Reincarnating RL":

See the discussion above.

## On "QDagger":

Strengths: It achieves impressive results in the tested settings.

Weaknesses: Given the seeming lack of originality of the algorithm, it is difficult to say that the contribution itself is highly significant.

# TL;DR

I don't think the large amounts of posturing in the paper is particularly helpful to the broader community; however, the core algorithm and its results are very well-done and clean enough to warrant an accept.

---

> ### Author Response · Authors · 2022-08-01
> **Author Response (Part 1): Added discussion on reproducibility and generalizability, Clarifications for democratization**
>
> We thank the reviewer for their constructive feedback. Main changes in revision, $\textcolor{red}{shown\ in\ red}$, are listed in “Summary of Changes” comment. We first address concerns related to limitations and questions raised by reviewer.
>
> > **This paradigm introduces its own host of problems .. "reincarnation from two distinct policies that perform similarly results in different performance trends". Thus, it seems as though it may be very easy to overfit to specific implementations and seeds of the pre-trained agents.**
>
> We’d like to clarify that the two distinct policies (Figure 10), mentioned in Line 336, were obtained through different agents and do not correspond to different random seeds: (1) Training DQN (Adam) from scratch for 400M frames, (2) Fine-tuning DQN(Nature), trained for 200M frames, with Adam for 20M frames. These two policies have similar IQM performance but different behaviors, which results in different reincarnation performance. Furthermore, reported results are aggregated across 3 random seeds of the pretrained policy for each of the 10 games, which shows the robustness to random seeds.
>
> We expect algorithmic rankings to typically be preserved with these two teachers despite resulting in different reincarnation performance. In fact, we empirically verified this by running an experiment to evaluate the top 3 methods in our work (QDagger, kickstarting, pretraining) using the fine-tuned Nature DQN teacher and found that the ranking of the methods does remain unchanged (**Figure A.11**). We have added this discussion in the **Appendix A.2**.
>
> > **The precise details of generation of teacher policy / data become important for reproducibility; adding on to the challenge of reproducing work from scratch.**
>
> We acknowledge that *reproducibility from scratch* is challenging and have added a discussion in **Appendix A.2**. The “reproducibility from scratch” issue also exists in NLP and computed vision, where existing pretrained models (e.g., GPT-3) are rarely, if ever, reproduced / re-trained from scratch but almost always used as-is. Despite this, “pretraining-and-fine-tuning” is a dominant paradigm in NLP and vision, and we believe this issue should not prevent us from doing research in reincarnating RL.
>
> Furthermore, reproducing existing computational work, which could be more expensive than training tabula rasa agents, beats the purpose of using reincarnating RL. Instead, reincarnating RL research would build on open-sourced prior computational work on a given benchmark. For example, to allow others to use the reincarnation setup in our work, we have already open-sourced [DQN model checkpoints and replay buffers ](https://console.cloud.google.com/storage/browser/rl_checkpoints) in the original submission.
>
> > **The limitations on reproducibility is problematic since science is about repeatedly testable hypotheses.**
>
> The testable hypothesis in reincarnating RL research would be about whether a given algorithm is better than another algorithm given the same prior computational work. For example, one hypothesis could be whether QDagger is a better algorithm than kickstarting, which can be repeatedly tested, for example, by rerunning PVRL experiments on ALE.
>
> > **I'm not sure if I agree with "reincarnating RL" as being a superior approach for democratizing RL research .. environments such as Minatar democratize research and allow for cheaper testing of tabula rasa RL.**
>
> We did not claim superiority of reincarnating RL over other approaches for democratizing RL research. That said, while it is worthwhile to do RL research on small-scale problems, it is also important to be able to investigate RL on more complex problems as otherwise we’d run the risk of overfitting to small-scale problems (for example, it is unclear whether better performing methods on MinAtar would perform better on more complex tasks like Starcraft or MineCraft?). However, research on large-scale RL domains is mostly accessible to researchers within compute-rich labs. Reincarnating RL would allow us to also tackle larger-scale domains without requiring excessive computational resources.
>
> A majority of reincarnating RL research would be about developing broadly applicable reincarnation algorithms as opposed to achieving higher scores. As pointed by #R4, this is important as a common use of RL algorithms in real-world settings will likely be in scenarios where prior computational work is available (previous data, policies etc.). For example, there is an [ongoing NeurIPS competition](https://minerl.io/basalt/#changes) on MineCraft where existing pretrained policies are provided as RL from scratch fails to get almost any reward [[Baker et al.](https://arxiv.org/abs/2206.11795)].

---

> > ### Author Response · Authors · 2022-08-01
> > **Author Response (Part 2): Prior reincarnation efforts were ad hoc, QDagger's significance, and Addressing questions.**
> >
> > > **Plenty of work that would fall into reincarnating RL, as seen in "Related Work". .. not sure reincarnating RL could be considered an original contribution**.
> >
> > As we mention in the paper, while several large-scale efforts have used reincarnation, it has typically been done in an ad-hoc way and has limited applicability. Existing approaches such as PBT (Starcraft) and Net2Net (Dota) can not be applied for reincarnating RL when switching to arbitrary architectures or from a policy to a value function. Similarly, behavior cloning (Rubik’s cube) is only useful for policy to policy transfer and is inadequate for the PVRL setting.
> >
> > Our formalization is arguing for studying reincarnating RL as a research problem and developing broadly applicable reincarnation approaches. For example, policy to value reincarnation setting studied in this work enable us to apply reincarnation to problems where prior reincarnation approaches are not applicable including (1) transferring an existing DQN policy to a Impala-CNN Rainbow agent in ALE, and (2) transferring Perciatelli (a distributed QR-DQN agent with MLP architecture) to a recurrent R2D2 agent in BLE.
> >
> > > **Given the seeming lack of originality of QDagger, it is difficult to say that the contribution itself is highly significant.**
> >
> > While QDagger is a simple algorithm, it works particularly well for PVRL and its significance should take into account that it allows us to efficiently apply reincarnation to problem settings where prior reincarnation approaches are not applicable (research workflow experiments on ALE and BLE). Our intention is for QDagger to serve as a starting point for future work in this area.
> >
> > > **What example settings (beyond BLE) are you thinking of when it comes to "real-world RL adoption"?  In what real-world scenarios would a broader community improving existing trained agents be helpful?**
> >
> > Any real-world setting where RL is currently deployed in practice and tabula rasa re-training is computationally / data expensive is a good candidate for adoption of reincarnating RL due to existence of prior computational work (such as existing deployed policies or data) and to tackle the challenge of being able to improve upon the existing deployed policy (which is often the case).
> >
> > RL problems where improving performance leads to real-world impact, such as chip design [1], tokamak control [2], compiler optimization [3], etc. are scenarios where continually improving existing trained agents might be helpful. We have added this point in the introduction.
> >
> > > **How do you imagine reincarnating RL research looks in practice?**
> >
> > We speculate that research in reincarnating RL can branch out in two directions:
> >
> > - **Standardized benchmarks with open-sourced computational work** (e.g., fixed teachers): This research would mostly focus on the development of reincarnating RL approaches. Akin to NLP and computer vision, where typically a small set of pretrained models are used in “pretraining and fine-tuning” research, this research can also possibly converge to a small set of open-sourced computational work (e.g., pre-trained teachers) on a given benchmark, for example, the teacher DQN agents and data we released on Atari.
> >
> > - **Real-world domains**:  Since obtaining higher performance has real-world impact in such domains, it incentivizes the community to build on current “state-of-the-art” agents and try to improve their performance. This would also likely benefit from methods' research on standardized benchmarks.
> >
> > > **How is a real-world practitioner supposed to use the learnings from this line of work on a new task?**
> >
> > The generalizable findings from reincarnating RL research would be about comparing algorithmic efficacy given access to same computational work (e.g., policies) on a specific task. As such, practitioners can use these findings to try to improve on an existing deployed RL policy (as opposed to being restricted to running tabula rasa RL). For example, this work shows that QDagger is effective for training a value-based agent given access to an existing policy on ALE and as such, QDagger can be tried in new tasks where we have an existing policy (e.g., BLE where we had the Perciatelli teacher).
> >
> > ----------------------------
> > *We hope that most of the reviewer’s concerns have been addressed and, if so, they would reconsider their assessment. We’d be more than happy to engage in further discussions.*

---

> > > ### Comment · Reviewer_jTCn · 2022-08-07
> > > **Response to Rebuttals**
> > >
> > > Thank you for the very well-done and extremely thorough response!
> > >
> > > My concerns have been addressed and I have updated the score.

---

### Official Review · Reviewer_kCKQ · 2022-07-15

**Rating:** 7
**Confidence:** 3
**Soundness:** 4 excellent
**Presentation:** 4 excellent
**Contribution:** 2 fair

**Summary:**

The authors investigate reincarnating RL, which aims to utilize prior policies to train new agents rather than training RL agents from scratch. This paper formalizes the reincarnation setting. They demonstrate that existing approaches fail in this setting, and their method, n-step QDagger, they are able to outperform prior approaches.

**Questions:**

1. Can the authors justify reincarnation in the most common RL settings? Also how should evaluations be reported in this case? Since reincarnating can lead to improved performance, but also has a dependence on an already trained teacher.

**Limitations:**

The work sufficiently addresses limitations and societal impacts.

**Strengths And Weaknesses:**

## Strengths
1. The results demonstrate that n-step QDagger is able to achieve strong performance (match, beat a fully trained DQN with far fewer steps in Atari, train faster than tabular rasa in humanoid:run, and outperform tabular rasa agents in BLE).
2. They paper includes numerous baselines and shows hyperparameter sweeps over prior agents and demonstrate pitfalls of these agents.
3. The paper felt quite lucid to read and typically straightforwards to follow. The bulleted list under evaluation was a little dense to read, not sure if it's easy or possible, but a better visualization of each method might make it easier to parse.

## Weaknesses
1. The main purpose of the paper is to formalize 'reincarnation.' Although I appreciate the extensive baselines and environments tested in the paper, it feels as if this work is formalizing a process that is in some ways well known. This process is valid; however, as noted by the authors already well documented in the RL community e.g. starcraft, dota, robotics for Rubik's cube solving.
2.  For a lot of RL tasks: single agent RL (atari, mujoco, car racing), robotics (Habitat, Sawyer arms, etc.), multi-agent RL (Hanabi, SMAC, MPE, etc), the training time is typically much shorter e.g. on the order of a dozen hours to a few days per policy. With good engineering, we can also reduce training times. For example, with a well implemented Ape-X or R2D2, we can significantly reduce RL training times -- these also can still be single machine 1 GPU training runs. As a result, in the context of focuses in the broader RL community, I'm not entirely sure how impactful/how big of a role reincarnation will play.
3. An issue of reincarnating could be related to diversity. A goal in reinforcement learning is to find diverse policies. Although this is not directly related to this work, reincarnating might not be able to fall into the workflow of training diverse policies.
4. I'm not entirely sure what teacher the paper uses to train the reincarnated agents for BLE? My impression is they're using the weights from Perciatelli et al. In this case it feels like it'd make sense that reincarnated outperforms other variants, as reincarnated bootstraps off of a strong policy that has seen a much wider space then it is basically fine tuned to the BLE environment via reincarnation. This also explains why the fine-tuned Perciatelli model performs well in the BLE environment.

## Minor
- I think the claim "JSRL does not improve performance compared to tabula rasa DQN and even hurts performance with a large number of teacher roll-in steps" feels a little strong, given that it only hurts performance when $n=1$. However, I don't think using multi-step = 1 is very common particularly in atari.
- I'm not entirely sure if the reference to Panel 3 in Figure 1 on line 251 is correct? Similarly, it would be helpful to reference the appropriate figure for the discussion in the second paragraph of sec. 5 (Fig A. 13, A. 14 I believe).

---

> ### Author Response · Authors · 2022-08-01
> **Author response (Part 1): Prior reincarnation efforts were ad hoc, justification for studying reincarnation, added discussion on evaluation**
>
> We thank the reviewer for constructive feedback. Main changes in revision, $\textcolor{red}{shown\ in\ red}$, are listed in “Summary of Changes” comment. We address reviewer's concerns below.
>
> > **The purpose of the paper is to formalize 'reincarnation.' ..  this work is formalizing a process .. as noted by the authors already well documented e.g. starcraft, dota, Rubik's cube.**
>
> While several large-scale efforts have used reincarnation, it has typically been done in an ad-hoc way and has limited applicability. Existing approaches such as PBT (Starcraft) and Net2Net (Dota) can not be applied for reincarnating RL when switching to arbitrary architectures or from a policy to a value function. Similarly, behavior cloning (Rubik’s cube) is only useful for policy to policy transfer and is inadequate for the PVRL setting.
>
> Our formalization is arguing for studying reincarnating RL as a research problem in its own right and developing broadly applicable reincarnation approaches. For example, policy to value reincarnation setting studied in this work enabled us to apply reincarnation to problems where prior reincarnation approaches are not applicable including (1) transferring an existing DQN policy to a Impala-CNN Rainbow agent in ALE, and (2) transferring Perciatelli (a distributed QR-DQN agent with MLP architecture) to a recurrent R2D6 agent in BLE.
>
> > **With a well implemented Ape-X or R2D2, we can significantly reduce RL training times -- role of reincarnation?**
>
> Distributed RL training complements reincarnation, rather than making it unnecessary. For example, training an agent on challenging domains, such as Starcraft, takes a long time (several weeks / months) even with distributed training with 1000s of actors.  As such, we can easily train reincarnated agents with distributed training, as we did on BLE (including a R2D2 variant).
>
> > **Current RL tasks .. training time is on the order of hours to days .. justify reincarnation in common RL settings?**
>
> As pointed by #R4 (mLNV), a common use of RL in the real-world will likely be in scenarios where prior computational work is available (e.g., previously-learned policies), which is evident from the prior large-scale reincarnation efforts, including applications such as chip design [1]. Thus, it is important to develop general-purpose reincarnation algorithms as opposed to prior ad-hoc solutions. Repurposing existing benchmarks for reincarnating RL, akin to how we use ALE in this work, can serve as testbeds for developing such reincarnation algorithms.
>
> Furthermore, reincarnating RL enables researchers with limited compute to tackle larger-scale RL problems, currently accessible only to compute-rich labs, than if they were using tabula rasa RL. For example, there is an [ongoing NeurIPS competition](https://minerl.io/basalt/#changes) on MineCraft where existing pretrained policies are provided as RL from scratch fails to get almost any reward [[Baker et al.](https://arxiv.org/abs/2206.11795)]. While it is worthwhile to do RL research on small-scale problems, investigating RL on large-scale problems helps avoid overfitting to such small-scale problems.
>
>
> > **How should evaluations be reported in reincarnating RL?**
>
> We acknowledge that reproducibility and fair evaluations in reincarnating RL is an important concern and have added a detailed discussion in **Appendix A.2**.
>
> While performance in reincarnating RL depend on prior computational work (e.g, teacher in PVRL), this is analogous to how fine-tuning results in NLP / computer vision depend on the pretrained models (e.g., using BERT vs GPT-3).  Fairly comparing reincarnation approaches requires using the exactly same computational work. Indeed, to allow others to use the same reincarnation setup as our work, we open-sourced
> [DQN model checkpoints and replay buffer](https://console.cloud.google.com/storage/browser/rl_checkpoints) in the original submission.
>
> The performance ranking of reincarnation algorithms is likely to remain consistent across different teachers on a benchmark. In fact, we empirically verified this for the PVRL setting, where we find that while using two different teacher policies, namely DQN(Adam) vs  a fine-tuned Nature DQN, leads to different performance trends but the ranking of PVRL algorithms remain consistent: QDagger  > Kickstarting > RL Pre-training (see Figure A.11).
>
> > **What teacher the paper uses to train the reincarnated agents for BLE?**
>
> We acknowledge this oversight and have rephrased the lines 284-285 to clarify this and added experimental details about BLE in Appendix A.5.3. Perciatelli is the name of the RL agent (QR-DQN with a MLP architecture) trained by Bellemare et al in their Nature paper using a distributed RL setup and released with the BLE as a starter agent.
>
> The BLE experiments show the benefit of reincarnating RL workflow, in which we are able to utilize the already released Perciatelli agent, as opposed to throwing it away as done by tabula rasa RL.

---

> > ### Author Response · Authors · 2022-08-01
> > **Author Response (Part 2): Addressing other / minor concerns**
> >
> > > **An issue of reincarnating could be related to diversity. A goal in reinforcement learning is to find diverse policies. Although this is not directly related to this work, reincarnating might not be able to fall into the workflow of training diverse policies.**
> >
> > This is an interesting point for discussion. Akin to tabula rasa RL, diversity-inducing objectives (e.g., entropy maximization) can also be added during the RL phase for a reincarnated agent. Thus, while reusing existing computational work (e.g., learned policies / data) can bias the learned policy towards existing behaviors, this can possibly be overcome with sufficient exploration. We’d be curious to hear from the reviewer whether they think that diversity challenges in reincarnation are fundamentally different from tabula rasa RL.
> >
> > > **I think the claim "JSRL does not improve performance compared to tabula rasa DQN and even hurts performance with a large number of teacher roll-in steps" feels a little strong, given that it only hurts performance when n=1. However, I don't think using multi-step = 1 is very common particularly in atari.**
> >
> > Thanks for pointing this out. We mistakenly referenced Figure 3 (right) instead of Figure 3 (left) for JSRL results and have fixed this in the revision.
> >
> > In Figure 3 (left), we see a big drop in performance with teacher roll-in steps set to 5000, even when using n-step=3. JSRL performance is comparable or much lower relative to the tabula-rasa DQN, which receives an IQM score of 0.39.
> >
> > > **I'm not entirely sure if the reference to Panel 3 in Figure 1 on line 251 is correct? Similarly, it would be helpful to reference the appropriate figure for the discussion in the second paragraph of sec. 5 (Fig A. 13, A. 14 I believe).**
> >
> > The reference to Panel 3 in Figure 1 is correct. Panel 3 shows the performance comparison between the tabula rasa Impala-CNN Rainbow (sky-blue), reincarnated Impala-CNN Rainbow (pink) and further fine-tuning the existing DQN agent (yellow).
> >
> >
> >
> >
> > *References*:
> >
> > [1] Mirhoseini, Azalia, et al. "A graph placement methodology for fast chip design." Nature 594.7862 (2021): 207-212.
> >
> > ------------------
> > *We hope that most of the reviewer’s concerns have been addressed and, if so, they would reconsider their assessment. We’d be happy to engage in further discussions.*

---

> ### Comment · Reviewer_kCKQ · 2022-08-09
> **Rebuttal Response**
>
> Apologies for the delay in reply. I appreciate the thoughtful response.
>
> My concerns have been addressed and updated my score.

---

> ### Public Comment · ~Roy_Fine1 · 2023-01-19
> **Thanks**
>
> Thanks for sharing your thoughts. Check out this website at https://writinguniverse.com/essay-types/persuasive-essays if you want to write an essay but are unsure of how to accomplish it. With the aid of our website, you may successfully complete your essay assignment on time.

---

### Author Response · Authors · 2022-08-02
**Thanks to reviewers & Summary of Changes**

We thank all the reviewers, $R1$ (kCKQ), $R2$ (jTCn), $R3$ (VM1W), and $R4$ (mLnV), for their valuable feedback! All reviewers found the paper to be well-written and presented, with sound results and extensive experiments and are in favor of accepting the paper. This work also generated a lot of questions and discussion from the reviewers, who mentioned that reincarnating RL would be of interest to various researchers.

Based on reviewers’ comments, we have **revised the paper** to add discussion and a couple of new experiments, $\textcolor{red}{shown\ in\ red}$, with clarifications (marked in $\textcolor{brown}{brown}$).  We summarize the changes below:

- *Reproducibility and Comparisons in Reincarnating RL*: Added detailed discussion in **Appendix A.2** [$R1, R2, R4$]
- Highlighted that prior reincarnation efforts were *ad hoc* and have limited applicability [$R1, R2$]
- *Generalizability in Reincarnating RL*. Added discussion in Appendix A.2 and results in **Figure A.11** showing that ranking of PVRL algorithms remain consistent across two different teachers [$R1, R2$]
- *Experimental details about BLE* for self-containedness in **Appendix A.5.2** [$R1, R3$]
- Clearly framing reincarnating RL as a class of problem settings, with examples in Appendix A.6 [$R4$]
- Additional *ablations* in **Appendix A.7** showing that QDagger works well without offline replay [$R3$] and impact of QDagger in online phase [$R4$]
- Included takeaway from n-step experiments and clarifications about humanoid-run [$R3$]

**Update**:  Based on our changes and response, reviewers $R1$, $R2$ and $R4$ acknowledged that their concerns have been addressed and updated their assessment to recommend *accept*, *accept* and *strong accept* respectively.

---

### Meta-Review · Area_Chair_uHXC · 2022-08-24

**Recommendation:** Accept
**Confidence:** Certain

**Metareview:**

This paper proposes a novel method for transferring prior policies across design and system changes to improve the sample efficiency of RL algorithms, which could ultimately help unlock RL for real-world use cases.
There was an active discussion across the reviewing process in which the authors managed to address the concern of the reviewers, leading to updated scores.

Based on the contributions of the paper and highly positive final reviews I recommend the paper for acceptance.


**Award:**

No

---

### Decision · Program_Chairs · 2022-09-14

Accept